# Chinese News Text Classification Method via Key Feature Enhancement

**Bin Ge, Chunhui He \*, Hao Xu, Jibing Wu and Jiuyang Tang**

Laboratory for Big Data and Decision, National University of Defense Technology, Changsha 410073, China
\* Correspondence: xtuhch@163.com; Tel.: +86-13786206971

**Featured Application: This method can be applied to Chinese news text classification task.**

**Abstract:** (1) Background: Chinese news text is a popular form of media communication, which can be seen everywhere in China. Chinese news text classification is an important direction in natural language processing (NLP). How to use high-quality text classification technology to help humans to efficiently organize and manage the massive amount of web news is an urgent problem to be solved. It is noted that the existing deep learning methods rely on a large-scale tagged corpus for news text classification tasks and this model is poorly interpretable because the size is large. (2) Methods: To solve the above problems, this paper proposes a Chinese news text classification method based on key feature enhancement named KFE-CNN. It can effectively expand the semantic information of key features to enhance sample data and then combine the zero–one binary vector representation to transform text features into binary vectors and input them into CNN model for training and implementation, thus improving the interpretability of the model and effectively compressing the size of the model. (3) Results: The experimental results show that our method can significantly improve the overall performance of the model and the average accuracy and $F_1$-score of the THUCNews subset of the public dataset reached 97.84% and 98%. (4) Conclusions: this fully proved the effectiveness of the KFE-CNN method for the Chinese news text classification task and it also fully demonstrates that key feature enhancement can improve classification performance.

**Keywords:** key feature; text classification; data augmentation; KFE-CNN; neural network

## 1. Introduction

Web news refers to news documents formed with a network as the carrier. It is different from traditional news content and it brings a brand-new experience to users in terms of transmission channels and reading forms. It can orderly integrate unordered content and greatly compress the amount of information, allowing the user to obtain the most valuable information in the shortest time. In the information world, web news as a mainstream method of information dissemination can be seen everywhere in daily life and a large amount of web news is spread on the internet every day. How to use high-quality text classification technology to help humans to efficiently organize and manage the massive amount of web news is an urgent problem to be solved.

Studies have shown that combining high-quality text classification methods can form effective organizational and management capabilities for mass news texts. This popular approach uses feature engineering combined with traditional machine learning methods to complete text classification simply and effectively, but the overall performance is not high. In most scenarios, it is a scientific and effective solution to realize the automatic classification of web news by building a large-scale news annotation dataset and combining deep learning algorithms to train high-quality text classification models [1]. It is worth noting that although such schemes can accurately solve the problem of automatic classification of news text, there are three problems that cannot be ignored: (1) The cost of building a large-scale

labeled dataset is extremely high. As we all know, high-quality labeled data is the core driving force for the development of artificial intelligence, which contributes to improving the performance of text classification algorithms. However, in some low-resource scenarios, it is impossible to completely rely on manual methods to achieve large-scale unlabeled news data labeling tasks. (2) The underlying deep learning method that uses a pre-trained language model to represent features is poorly interpretable. Even deep learning methods that incorporate pre-trained language models have achieved impressive results in feature representation and text classification tasks. The pre-trained language model is a black-box model with a large number of parameters and poor interpretability in the form of distributed vector outputs. In particular, the text classification method combined with transfer learning theory, using the prior knowledge learned by the pre-trained language model, interferes with the overall interpretability of the model in the process of parameter transfer, feature transfer and fine-tuning. (3) Most deep learning models are large in size and low in computing performance. After the existing deep learning model is combined with the pre-trained language model, many additional parameters are introduced to make the model bulky and the vector form corresponding to the language model is a distributed dense vector composed of floating-point values, which reduces the calculation performance of the model. The comparative analysis results of different methods are shown in Table A1 in Appendix A.

In summary, it is of great significance to improve existing methods from the aspects of reducing the cost of large-scale labeling datasets, improving the interpretability of models, compressing models' sizes and improving classification performance [2]. By analyzing the deficiencies of existing methods, this paper proposes a key feature-enhanced Chinese news text classification method named KFE-CNN. The KFE-CNN method can effectively reduce the cost of labeling large-scale datasets and achieve high-performance text classification goals. In addition, the KFE-CNN method uses a zero–one binary vector to represent features, which can effectively realize feature quantization and model volume compression. The details of the KFE-CNN method are described below. First, some key features are extracted from the original labeled data through a key feature extraction algorithm. Then, through feature semantic association technology, the selected key features are semantically extended to obtain more semantically-related extended features. Finally, the above-mentioned original key features and extended features are concatenated together as an explicit input model of classification features to complete training and inference. It is worth noting that all features of the KFE-CNN method are converted into dense vectors in a zero–one binary vector manner using feature engineering technology, so that the bottom layer does not need to use a pre-trained language model for feature representation, thereby improving the overall interpretability of the model. In addition, by using the feature expression advantages of the zero–one binary vector, the model size can also be compressed indirectly. Part of the original manual annotation data is expanded through the key feature semantic extension technology, so as to automatically build a high-quality annotation dataset, which can effectively reduce the dependence on large-scale original manual annotation data. The core contributions of this paper are as following:

(i)   A key feature-enhanced Chinese news text classification method is proposed, which can complete data enhancement based on part of the original labeled data and construct a high-quality labeled sample set to solve the problem of large-scale Chinese news text classification;

(ii)  We improve the overall classification performance and interpretability of the model through the feature engineering technology with key feature extraction and semantic information expansion and complete the compression of the model size through zero–one binary vector integer quantization technology;

(iii) We discuss the necessity of sampling enhancement in the Chinese news classification task and also provide a simple and efficient solution.

## 2. Related Work

Text classification has a long history in the field of natural language processing (NLP) [3]. The early research was mainly based on expert rules from the 1980s. It was developed to use knowledge engineering to build an expert system to realize the text classification. Until the early 21st century, with the explosive growth of the number of internet texts and the maturity of machine learning theory, a set of solutions to large-scale text classification problems, based on feature engineering combined with classification models, was gradually formed. Feature engineering plays an extremely important role in machine learning methods. Generally speaking, a machine learning problem is the process of converting data into information and then refining it into knowledge. The role of feature engineering [4] is mainly to extract information from data, while the role of the classifier is to transform the extracted information into knowledge. Feature engineering mainly covers three parts: text preprocessing, feature extraction and feature representation. The ultimate goal is to convert text into a computer-understandable vector form. The traditional text representation method is mainly represented by the bag of words [5] or the vector space model [6]. The disadvantages of these methods are that the text context is ignored and the features are independent of each other. In order to improve the above shortcomings, Hinton [7] proposed a distributed representation theory of features, in which the main idea is to express each word as an n-dimensional dense real number vector. In the early stage, the one-hot vector [8] was mainly used for representation. With the development of technology, a series of excellent distributed representation technologies such as word2vec [9], Glove [10], BERT [11] and GPT-X [12] were successively proposed, which greatly promoted the development of feature representation technology.

The current mainstream classification models are mainly divided into two categories: traditional statistical learning methods and deep learning methods. The most common statistical learning models for text classification tasks are Support Vector Machine (SVM) [13] and Naive Bayesian (NB) [14] models. The advantage of this type of model is that it is simple and effective, but the disadvantage is that it needs to use manual feature engineering to extract feature vectors, which is time-consuming and has poor scalability. With the development of deep learning technology, scholars have proposed advanced classification models such as FastText [15], CNN [16], RNN [17] and their series of variants and have achieved good classification results. The advantage of the deep learning model is that it can accurately predict the text category by combining the distributed word vector representation technology with the deep neural network. However, the disadvantage is that large-scale labeled text datasets need to be manually annotated to train high-quality classifiers. In order to improve the model, the literature [18–20] has proposed a variety of effective data enhancement methods to improve the performance of text classification through content compression or content expansion.

One must comprehensively consider the huge advantages of feature engineering and deep learning methods as well as data enhancement techniques in text classification tasks. This paper proposes a key feature-enhanced Chinese news text classification method through in-depth study of the advantages of the three major technologies. It can extract key features based on part of the original annotation data and complete semantic extension to achieve data enhancement to automatically expand the annotation samples. Finally, combination with deep convolutional neural network model efficiently realizes the automatic classification of news texts.

## 3. Key Feature-Enhanced Chinese News Text Classification Method

By fully integrating the advantages of feature engineering and deep learning methods, a key feature-enhanced news text classification model is proposed. The model first performs data enhancement processing on the original text content through two methods of key feature extraction and key feature semantic extension and then converts all the enhanced features into the form of zero–one binary integer vectors. Then, it inputs the obtained feature vectors into the CNN model, completes the training and inference and, finally,

gives the classification and probability through the SoftMax classifier. The flowchart of the KFE-CNN method is shown in Figure 1.

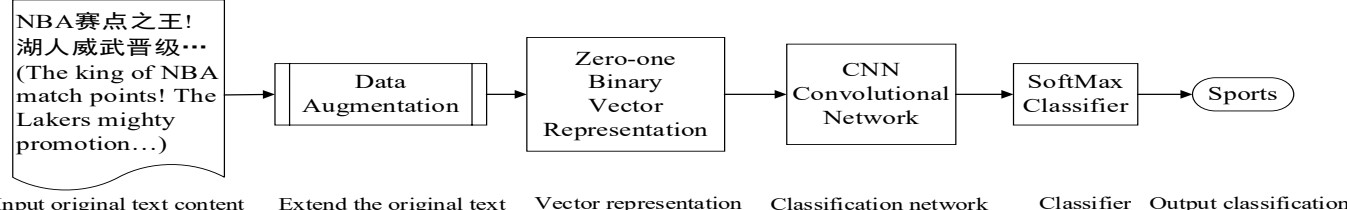

**Figure 1.** Flowchart of the KFE-CNN classification model.

### 3.1. Data Augmentation

As a core innovation of this research, the data augmentation module has played a huge role in improving the performance of the model. The expansion of all sample data in long text data will inevitably introduce some noise, which interferes with the classification results and will also greatly increase the time-consuming of sample data expansion. Considering the above problems, there are two major challenges in the data augmentation stage. The first is determining which kind of data samples need to be augmented to be identified. The second is determining which method should be used for data expansion for those samples that need data enhancement to have a better effect. In solving the above two problems through statistical analysis, it is found that most of the original samples in the training set in the experimental data are long texts with a length of more than 500 characters, a small number of samples are short texts with a length of less than 300 characters and the length of the rest samples is between 300 and 500 characters. According to the above statistical results, an efficient data expansion rule was designed. The data augmentation process is shown in Figure 2.

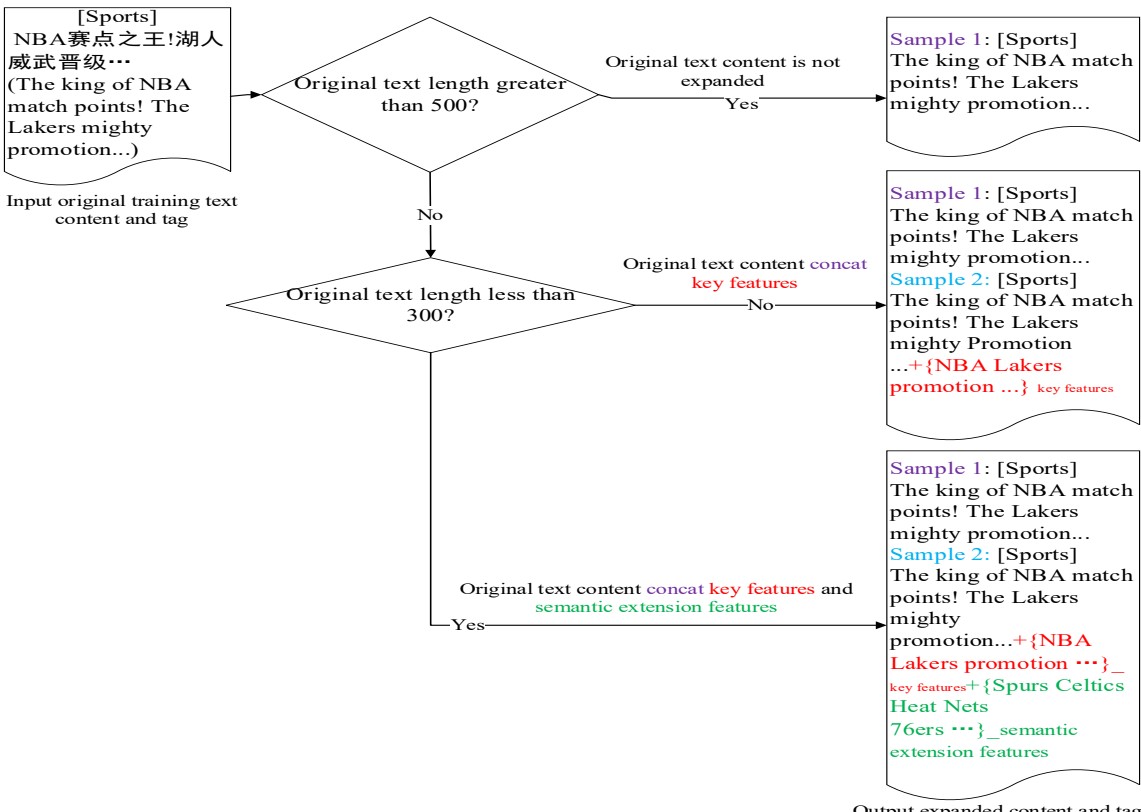

**Figure 2.** The flowchart of data augmentation processing for training and prediction samples.

Considering that the amount of information covered by the samples whose original text length exceeds 500 characters is sufficient, such samples do not require additional data expansion processing. For samples whose original text length is between 300 and 500 characters, the key features are directly extracted from the original text and then concatenated with the original text content as the newly generated content to complete the expansion of the sample. For samples whose original text length is less than 300 characters, the key features are first extracted from the original text and then the above key features are semantically extended from the external word vectors using the open source word vector calculation tool [21]. Finally, extracted key features and the externally expanded semantic features are concatenated with the original text content as the newly-generated content to complete the double expansion of the training samples. It should be noted that each original training sample with a length of less than 500 characters retains the original sample and the newly-generated extended sample after being expanded and then marks the corresponding label of the original sample to reconstruct two pieces of sample data, thus completing data augmentation for training samples. After the above data enhancement process, all sample data with a length of less than 500 characters in the training set can be effectively expanded. In addition, in order to form an experimental comparison, the validation set and test set in the extended dataset are consistent with the corresponding data samples in the original dataset without any data enhancement processing. Figure 2 shows the data enhancement process for the entire training set and the key feature extraction and feature semantic extension techniques in Figure 2 are described in detail in Sections 3.1.1 and 3.1.2.

### 3.1.1. Key Feature Extraction

In the Chinese text classification task, the classification model essentially extracts features based on the input text content and converts it into a vector form for training and inference. The discriminative signal of the proposed features has a direct impact on the model performance. Considering the above factors, the rank mode in the open source LAC2.0 lexical analysis tool [22] is used to analyze the importance of words in all original data samples whose content length is less than 500 characters. The entire rank model requires the input of the original content of an entire article and then returns three lists: word segmentation results, part-of-speech tagging results and the importance level corresponding to each word. This tool divides the importance of words into four distinct levels: 0, 1, 2 and 3. The higher level represents more importance for a word. Level 0 indicates redundant words in the document; common parts-of-speech are prepositions, punctuation marks, function words, etc. Level 1 indicates weakly-defined words in the document; common parts-of-speech are pronouns, conjunctions, particles, etc. Level 2 indicates strongly-qualified words in the document; common parts-of-speech are common nouns, location nouns, common verbs, etc. Level 3 indicates the core words in the document; common parts-of-speech are proper nouns, entity names, work names, etc. In the whole key feature extraction process, this method only classifies the words whose importance level is 3 in the original document as the final key features. All key feature extraction steps are shown in Figure 3.

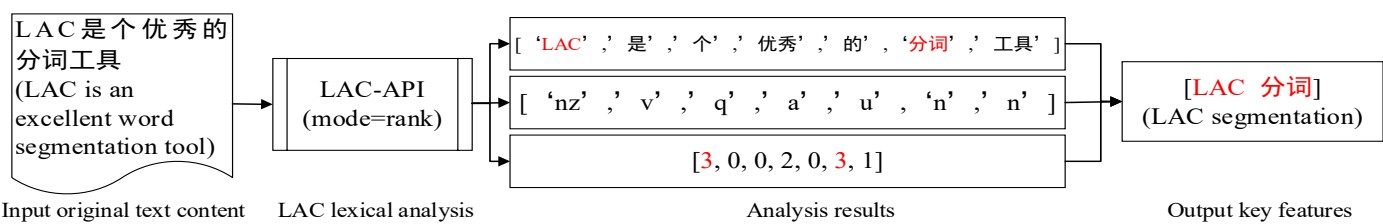

**Figure 3.** The flowchart of the key feature extraction process.

### 3.1.2. Key Feature Semantic Extensions

As can be seen from the flowchart in Figure 2, the entire key feature semantic extension only performs secondary data enhancement for samples whose original text length is less

than 300 and automatically skips this step if the original text length is not less than 300. The detailed semantic extension steps are as following:

Step 1: Input a sample whose original text length is less than 300 and initialize an empty semantic extension list SE, then go to Step 2;

Step 2: Use the key feature extraction algorithm in Section 3.1.1 to extract a list of all key feature words from the original text, then go to Step 3;

Step 3: Loop through the list of key feature words in Step 2, use each word as a query, input the large-scale word2vec word vector tool that is open source in the literature [21] and call the similar_by_word (query, 5) method to return the current query and the top-five most semantically-relevant feature words and their corresponding semantic similarity dictionary, T, then go to Step 4;

Step 4: If T is empty, go directly to Step 3 to start the expansion of the next query;

Step 5: If T is not empty, loop through the dictionary T and then add all the words in T with a semantic similarity value greater than S (in the experiment, S is set to 0.85) into the semantic expansion list SE, then go to Step 3 and start the next query extension;

Step 6: After traversing all queries, output the SE list as the final semantic extension features list.

The semantic extension of all key features can be realized by using the key feature semantic extension method in Steps (1) to (6). Then, this semantic extension feature is combined with the key features extracted and concatenated to the back of the original text to generate new training samples to achieve the final data enhancement. It is worth noting that the semantic similarity calculation method in Step 5 uses the cosine similarity to calculate the cosine distance between two k-dimensional vectors and the relevant calculation principle is shown in Formula (1):

$$S = \frac{\sum_{k=1}^{n}(x_k \times y_k)}{\sqrt{\sum_{k=1}^{n}(x_k)^2} \times \sqrt{\sum_{k=1}^{n}(y_k)^2}} \tag{1}$$

### 3.2. KFE-CNN Text Classification Method

Considering the advantages of convolutional networks in classification problems, CNN is selected as the skeleton network for the entire Chinese news text classification, which contains a total of seven layers from the input text to the end of the output category probability. The relevant process of the entire KFE-CNN text classification method is shown in Figure 4.

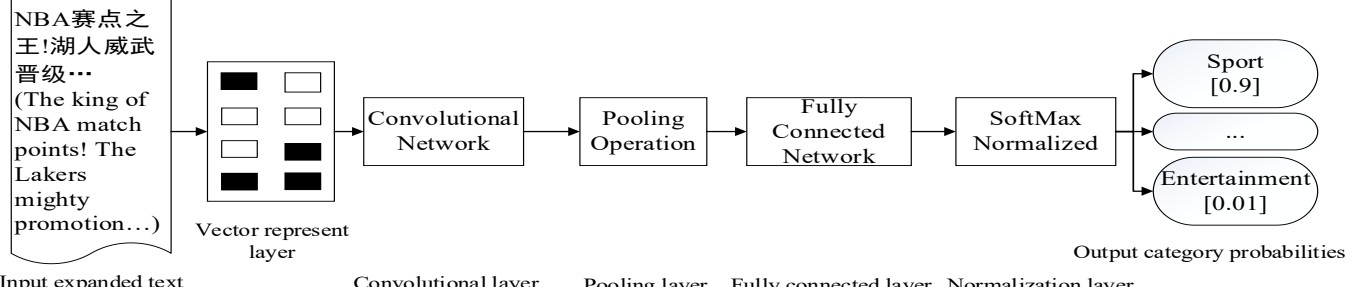

**Figure 4.** The structure of the KFE-CNN classification model.

According to Figure 4, it can be seen that the input layer is mainly used to input expanded text content. Then, in the vector representation layer, all the text content is represented as a zero–one binary distributed vector form. Next, the distributed vector is input into the convolution layer to realize the extraction of characteristic features. The character feature vector is input into the pooling layer and then the maximum pooling method is applied to complete the pooling operation for all feature maps. The processed feature vectors are then fed into a fully connected layer to map them to the sample label

space. Finally, the calculation results of the fully-connected layer are input into the normalization layer and the SoftMax function is used to output the normalized categories and their corresponding probability values. The relevant calculation of the normalization functions SoftMax is shown in Formula (2):

$$P(t_k) = \frac{e^{t_k}}{\sum_{k=1}^{N}(e^{t_k})} \tag{2}$$

where $P(t_k)$ represents the probability value corresponding to the $k$-th category, $t_k$ represents the $k$-th category and $N$ refers to the total number of categories. After the SoftMax normalization calculation step, the category corresponding to the maximum probability value can be output as the final text classification result.

## 4. Experiments

### 4.1. Datasets

The original Chinese dataset #DataSetA used in the experiment is constructed from a subset of THUCNews (https://pan.baidu.com/s/1vgnrvWBI-u4H16hIOJwXLA?pwd=th7s (accessed on 19 April 2023)). The entire dataset covers 10 categories, including #Sports, #Finance, #Real Estate, #Home Furnishing, #Education, #Technology, #Fashion, #Current Affairs, #Games and #Entertainment. The original experimental dataset contains 6500 (5000 for training, 500 for validation and 1000 for testing) samples for each category and a total number of 65,000 data samples. In addition, a total number of 18,259 new data samples (#Sports 659, #Finance 1336, #Real Estate 888, #Home Furnishing 4502, #Education 913, #Technology 1219, #Fashion 3393, #Current Affairs 2307, #Games 1932 and #Entertainment 1110) were expanded after the data enhancement processing step based on the original training samples of dataset #DataSetA and, finally, a new dataset, #DataSetB, was obtained, which contained a total number of 83,259 data samples (68,259 for training, 5000 for validation and 10,000 for testing) for all categories. It should be noted that #DataSetA is a completely balanced dataset. The training sample part of #DataSetB is a nearly-balanced dataset and the validation sample and test sample part in #DataSetB is exactly the same as #DataSetA. The statistics of relevant datasets are shown in Table 1.

**Table 1.** Statistical information derived from the experimental dataset.

| Dataset | #Training Samples | #Validation Samples | #Test Samples | #Sample Size | #Num_classes |
|---|---|---|---|---|---|
| #DataSetA (Original) | 50,000 (Original) | 5000 (Original) | 10,000 (Original) | 65,000 (Original) | 10 |
| #DataSetB (Enhanced) | **68,259** (Enhanced) | 5000 (Original) | 10,000 (Original) | **83,259** (Enhanced) | 10 |

The numbers without bold in Table 1 represent the original data samples and the bold numbers represent the enhanced sample datasets.

### 4.2. Evaluation Metrics

In the experimental stage, in order to fully verify the comprehensive performance of the model, in addition to the popular Precision, Recall and $F_1$-score, Accuracy indicators are also used to evaluate the model performance. The model uses cross entropy as the loss value to optimize the network. According to the confusion matrix shown in Table 2, the calculation steps of the above indicators are shown in Formulas (3)–(6).

$$Precison = TP/(TP + FP) \tag{3}$$

$$Recall = TP/(TP + FN) \tag{4}$$

$$F_1 = \frac{2 * Precison * Recall}{Precison + Recall} \tag{5}$$

$$Accuracy = (TP + TN)/(TP + TN + FP + FN) \tag{6}$$

**Table 2.** Structure of confusion matrix.

| Confusion Matrix | | System Results | |
| --- | --- | --- | --- |
| | | Positive (Erroneous) | Negative (Correct) |
| Gold Standard | Positive | TP (True Positive) | FN (False Negative) |
| | Negative | FP (False Positive) | TN (True Negative) |

The calculation steps of the cross entropy for the entire model of the loss function are shown in Formula (7):

$$Loss = CE(\bar{y}, y) = -\sum_{k=0}^{N-1} y_k log(\bar{y}_k) \tag{7}$$

where $N$ refers to the total number of categories, $\bar{y}$ represents the predicted probability distribution, $y$ represents the real probability distribution and $k$ represents the category dimension.

*4.3. Hyperparameter Settings for KFE-CNN Models*

After optimizing the core parameters of multiple rounds of experiments, a set of optimal hyper-parameter combinations was obtained for the experimental dataset. The relevant hyper-parameter settings of the entire KFE-CNN and CNN model are shown in Table 3.

**Table 3.** Hyper-parameter settings of the CNN and KFE-CNN model.

| Parameter Name | Parameter Value | Parameter Name | Parameter Value |
| --- | --- | --- | --- |
| Dropout_rate | 0.5 | Optimizer | Adam |
| Embedding_dim | 100 | Seq_length | 1200 |
| Num_filters | 256 | Num_classes | 10 |
| Kernel_size | 5 | Vocab_size | 15,000 |
| Hidden_dim | 128 | Learning_rate | $1 \times 10^{-3}$ |
| Batch_size | 64 | Num_epochs | 12 |

*4.4. Experimental Results and Analysis*

Experiments are conducted in order to reveal the comprehensive performance of the model from different aspects. Firstly, we conduct comparative experiments using eight different popular classification models on the original dataset #DataSetA. Then, at the macro level, three evaluation indicators (average accuracy, min-loss value and average $F_1$-score) are used to complete the final performance evaluation, so as to find the optimal classification model of #DataSetA. The relevant experimental results are shown in Table 4.

**Table 4.** Experimental results of different classification models on the original dataset #DataSetA-Test.

| Model Name | Avg Accuracy (%) | Min-Loss | Avg $F_1$-Score (%) |
| --- | --- | --- | --- |
| TF-IDF [23] | 89.26 | – | 85 |
| Naive Bayes [14] | 90.13 | – | 90 |
| SVM [13] | 92.39 | – | 91 |
| FastText [15] | 92.24 | – | 92 |
| LSTM [24] | 93.17 | 0.22 | 93 |
| RNN [17] | 94.49 | 0.21 | 94 |
| TextGCN [25] | 92.78 | 0.24 | 92 |
| CNN [16] | 96.26 | 0.13 | 96 |

According to the results in Table 4, it can be seen that the comprehensive performance of the CNN model is better than that of the other seven models in the eight groups of comparative experiments carried out on the original dataset #DataSetA-Test. Its average accuracy value can reach 96.26%, the min-loss value is 0.13 and the average $F_1$-score across all categories is 96%. The comprehensive performance of RNN is second, with an average $F_1$-score of 94% and the average $F_1$-score of other models is between 85% and 93%.

According to the above experimental results, we select the CNN skeleton model to build the KFE-CNN model to carry out further comparative experiments on the original dataset #DataSetA and the extended dataset #DataSetB and combine the micro-evaluation indicators between different categories to compute the comprehensive performance of different models. The relevant experimental results are shown in Table 5.

**Table 5.** Experimental results of the KFE-CNN model for different datasets (↑improvement, ↓decline).

| Dataset | Classification | Precision (%) | Recall (%) | $F_1$ (%) |
|---|---|---|---|---|
| #DataSetA (Original) CNN | Sports | 99 | 99 | 99 |
| | Finance | 95 | 99 | 97 |
| | Real Estate | 100 | 100 | 100 |
| | Home Furnishing | 99 | 87 | 92 |
| | Education | 88 | 95 | 91 |
| | Technology | 94 | 98 | 96 |
| | Fashion | 96 | 97 | 97 |
| | Current Affairs | 97 | 93 | 95 |
| | Games | 98 | 97 | 97 |
| | Entertainment | 98 | 97 | 98 |
| | Num_classes | Avg Accuracy (%) | Min-Loss | Avg $F_1$-score (%) |
| | 10 | 96.26 | 0.13 | 96 |
| **Dataset** | **Classification** | **Precision (%)** | **Recall (%)** | **$F_1$ (%)** |
| #DataSetB (Enhanced) **KFE-CNN** | Sports | 100 (↑1%) | 100 (↑1%) | 100 (↑1%) |
| | Finance | 98 (↑3%) | 99 | 98 (↑1%) |
| | Real Estate | 100 | 100 | 100 |
| | Home Furnishing | 97 (↓2%) | 93 (↑6%) | 95 (↑3%) |
| | Education | 96 (↑8%) | 97 (↑2%) | 96 (↑5%) |
| | Technology | 97 (↑3%) | 98 | 98 (↑2%) |
| | Fashion | 95 (↓2%) | 99 (↑2%) | 97 |
| | Current Affairs | 97 | 97 (↑4%) | 97 (↑2%) |
| | Games | 99 (↑1%) | 98 (↑1%) | 99 (↑2%) |
| | Entertainment | 99 (↑1%) | 98 (↑1%) | 99 (↑1%) |
| | Num_classes | Avg Accuracy (%) | Min-Loss | Avg $F_1$-score (%) |
| | 10 | **97.84 (↑1.58)** | **0.074 (↓0.056)** | **98 (↑2%)** |

According to the experimental results in Table 5, it can be seen that the performance of the KFE-CNN model for the enhanced dataset #DataSetB is better than that for the original dataset #DataSetA and a number of comprehensive evaluation indicators have been significantly improved. Compared with the original dataset #DataSetA from a macro perspective, the average accuracy of the KFE-CNN model for the enhanced dataset #DataSetB has increased by 1.58% to 97.84%, the min-loss value has decreased by 0.056 and the average $F_1$-score has increased by 2% to 98%. At the micro level, after the data enhancement processing, the comprehensive performance of the FKE-CNN model's evaluation indicators in different categories has been significantly improved. Among them, only the precision indicators of the two categories of #Home Furnishing and #Fashion have dropped by 2%, but they are all above 95%. There is a significant increase in all remaining categories, especially #Education, which sees an 8% improvement. Recall and $F_1$-score have different improvements in all categories or are the same as the original dataset. It is worth noting that the $F_1$-scores in the categories of #Education, #Home Furnishing, #Technology, #Current Affairs and #Games have increased by more than 2%, the $F_1$-scores in the categories of #Sports, #Finance, and

#Entertainment have increased by 1% and the $F_1$-scores are the same as for the original dataset in the categories of #Real Estate and #Fashion.

*4.5. Discussion*

In order to illustrate the advantages of this method from different aspects, an overview is mainly developed from the following four questions. Question 1: In the news text classification task, is it necessary to carry out data enhancement on the samples? According to the experimental results in Section 4.4, it can be found that data enhancement is very necessary, although data enhancement processing will introduce additional noise to a small number of samples and reduce the classification performance. However, in general, as long as the data enhancement strategy is good enough, the overall performance of the model can be significantly improved through data enhancement technology. Question 2: Is it better for the data augmentation strategy to augment only some samples or to augment all samples? In our experiments, it can be seen that for long text sample data, it is better to use data for enhancement only for samples whose original content length is less than 500. There are two main reasons: the first is that the time cost of partial sample data enhancement is lower than that of all sample data enhancement; the second is that we found, through comparative experiments, that after all sample data is enhanced, the overall classification performance of the model is lower than that of some samples after enhancement. The average $F_1$-score is 97%, but it is still slightly higher than the original CNN model without data enhancement. Question 3: The new sample dataset obtained by enhancing part of the sample data sees a significant improvement for the KFE-CNN model, but is it equally effective for other models? In order to answer this question, we select the RNN skeleton model and carry out comparative experiments on #DataSetB and find that the comprehensive performance of the KFE-RNN model on #DataSetB has also been significantly improved. Compared with the original RNN model, the average accuracy of KFE-RNN has increased by 1.7% to 96.19%, the min-loss value has dropped to 0.14 and the average $F_1$-score has increased by 2% to reach 96%, which fully demonstrates that some sample data enhancement can effectively improve the overall performance of related models. The relevant comparative experimental results for Question 2 and Question 3 are shown in Figure 5a,b.

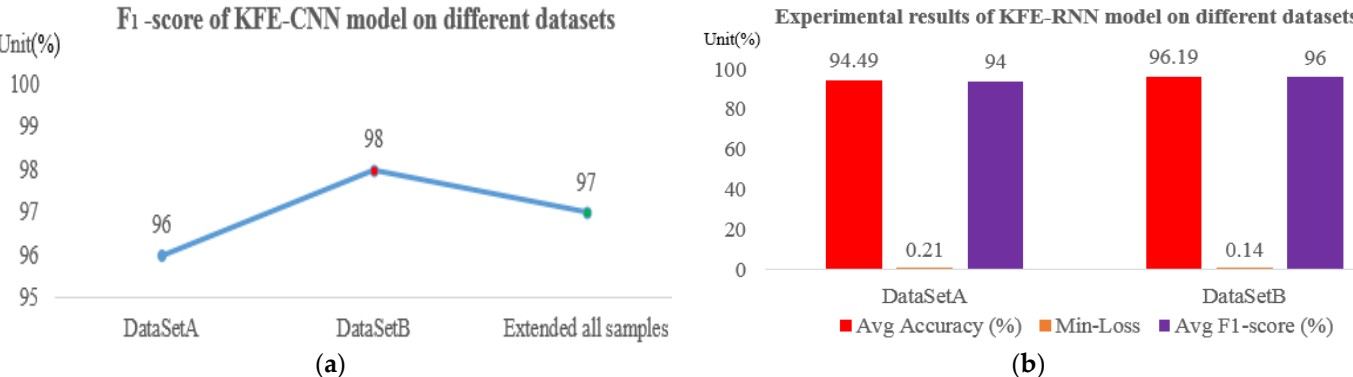

**Figure 5.** The experimental results of (**a**) KFE-CNN and (**b**) KFE-RNN models on different datasets.

Question 4: When using a deep learning model to solve the text classification task, will quantized features have a great impact on the overall performance and model size? Generally speaking, after the vectors are quantized, the accuracy of the model decreases due to the loss of precision. However, because the KFE-CNN method uses high-quality key feature semantic extension technology to expand many additional features, even if the quantization operation reduces the accuracy, the experimental results show that high-quality feature engineering can bring about a greater performance improvement. The overall performance of the method is not greatly reduced but has been improved. In addition, compared to the original CNN text classification method combined with the Bert pre-training model to complete the feature representation, the overall model size is more

than 100 MB. Because the KFE-CNN model uses zero–one binary integer quantization feature representation technology, there is no need to introduce an additional pre-trained language model to compress the overall model size to less than 10 MB.

**Shortcomings summary.** First of all, the KFE-CNN model has only been validated for Chinese text and has not been analyzed for feasibility in other languages. Secondly, our model currently performs well in the classification of news texts, but it has not been verified in other types of text. Thirdly, the current key feature enhancement module relies on LAC tools, but we could consider replacing them with analysis tools supporting other languages.

## 5. Conclusions and Future Work

News text classification is a popular direction in natural language processing. Existing deep learning methods rely on a large-scale annotated corpus in news text classification tasks and the interpretability of these models is poor. In order to improve the above phenomenon, a Chinese news text classification method with key feature enhancement is proposed. It effectively achieves data enhancement by extending the key features and then uses a zero–one binary vector to transform the text features into a distributed form and input them into the KFE-CNN model for training and implementation, further improving the interpretability of the model. The experimental results show that our method can not only greatly improve the overall performance of the model, but also reduce the dependence on large-scale manual labeling of data and compress the final checkpoint model size. In addition, for long text sample data, the experiment found that partial sample data enhancement is better than full sample enhancement and the enhanced sample data is universal to different classification models.

The KFE-CNN method is mainly verified on the public long text Chinese news dataset and it can be considered for extension to short-text classification tasks in the future. In addition, the current feature expansion mainly uses two methods of important word expansion and semantic similarity expansion to complete the enhancement of key features. In future research, we will continue to improve the KFE-CNN model to adapt to other languages and further establish experiments to explore its adaptability to other types of text. In addition, it is also possible to introduce an artificial intelligence content model similar to ChatGPT [26] to automatically generate relevant content based on key features to achieve feature enhancement and improve the overall classification performance of the model.

**Author Contributions:** Conceptualization, B.G. and C.H.; Data curation, C.H.; Formal analysis, C.H.; Funding acquisition, J.T.; Investigation, J.W. and J.T.; Methodology, C.H.; Project administration, B.G. and H.X.; Resources, B.G. and C.H.; Software, C.H.; Supervision, H.X. and J.W.; Validation, H.X. and J.W.; Visualization, B.G. and J.T.; Writing—original draft, B.G. and C.H.; Writing—review & editing, B.G. and C.H. All authors have read and agreed to the published version of the manuscript.

**Funding:** This research received the support of the National Natural Science Foundation of China (NSFC) via Grant No. 71971212.

**Institutional Review Board Statement:** Not applicable.

**Informed Consent Statement:** Not applicable.

**Data Availability Statement:** https://pan.baidu.com/s/1vgnrvWBI-u4H16hIOJwXLA?pwd=th7s.

**Acknowledgments:** We would like to thank THUCNews (http://thuctc.thunlp.org), a large-scale Chinese news text classification dataset open sourced by Tsinghua University, and the large-scale Chinese word vector (v-0.2.0) (https://ai.tencent.com/ailab/nlp/en/embedding.html), open sourced by Tencent AI Lab and the LAC (https://gitee.com/lixufeng90/lac). We also acknowledge the Chinese lexical analysis tool open sourced by Baidu, which helped us to carry out the relevant experiments in this paper. Thanks to all reviewers for their valuable revision suggestions.

**Conflicts of Interest:** The authors declare no conflict of interest.

## Appendix A

Table A1 shows the comparative analysis results for existing mainstream methods of text classification.

**Table A1.** Comparative analysis results of existing mainstream methods of text classification.

| Type of Methods | Large-Scale Manual Annotation Dataset | Pre-Training Language Model | Volume of Model | Classification Performance |
|---|---|---|---|---|
| Traditional classification model (TF-IDF/NB/SVM) | Independency | Not required | Smaller | Relatively low |
| Popular neural network model (Fast-Text/LSTM/RNN/TextGCN/CNN) | Dependency | Generally required | Large | Relatively high |
| Our method | Barely dependent | Not required | Smaller | Relatively high |

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
