# Peer review of "Chinese News Text Classification Method via Key Feature Enhancement"

_applsci, doi:10.3390/app13095399_

Round 1

Reviewer 1 Report

The following comments to be incorporate

-3.1.2 section steps write in proper format

-equation 3,4,5 metrics nomenclature write in proper format

-comparative analysis required of dataset with & without Data Augmentation by your proposed model with KFE-CNN

-Justify for taking different DataSetA & DataSetB, why only two and on which basis selected this 2 dataset, please enlight on this.

-in table 5 results ,how many samples taken for each class ,given total samples

Author Response

Response to the comments

Dear experts:

First of all, thank you very much for taking the time to help us review the paper and give professional and objective suggestions for revision. I am very happy that our work can be affirmed by you. Next, we will give a point-to-point revised reply to your comments, as follows:

Q1:-3.1.2 section steps write in proper format

A1: The revision has been completed. Please refer to the revised paper in Section 3.1.2.

Q2:-equation 3,4,5 metrics nomenclature write in proper format

A2: All equations have been updated.

Q3:-comparative analysis required of dataset with & without Data Augmentation by your proposed model with KFE-CNN

A3: Because KFE-CNN is a model based on CNN, its experiment on the original data set refers to CNN's experiment on the original data set. The difference between these two models lies in whether there is a data enhancement module. The main reason for using CNN model on the original data set is to keep consistent with CNN in Table 4.

Q4:-Justify for taking different DataSetA & DataSetB, why only two and on which basis selected this 2 dataset, please enlight on this.

A4: DatasetA (THUCNews subset) is an open multi-classification data set of Chinese news texts, which has certain authority and universality. DatasetB is an enhanced data set completely based on DatasetA through our key feature enhancement method. The two data sets are selected to carry out experiments, mainly considering the authority and universality of the data sets, as well as the data quality and the differences between the data sets, which can clearly reveal the classification performance and reliability of our method.

Q5:-in table 5 results, how many samples taken for each class, given total samples

A5: For the description of the data set, please refer to the description in Section 4.1 of the manuscript. The original Chinese dataset #DataSetA used in the experiment are constructed from a subset of THUCNews[1]. The entire dataset covers 10 categories including #Sports, #Finance, #Real Estate, #Home Furnishing, #Education, #Technology, #Fashion, #Current Affairs, #Games, and #Entertainment. The original experimental dataset contains 6,500 (5,000 for train, 500 for validation, and 1,000 for test) samples for each category, and a total number of 65,000 data samples. In addition, a total number of 18,259 new data samples (#Sports 659, #Finance 1,336, #Real Estate 888, #Home Furnishing 4,502, #Education 913, #Technology 1,219, #Fashion 3,393, #Current Affairs 2,307, #Games 1,932, and #Entertainment 1,110) were expanded after data enhancement processing based on the original Training Samples of dataset #DataSetA, and finally a new dataset #DataSetB was obtained, which contained a total number of 83,259 data samples (68,259 for train, 5,000 for validation, and 10,000 for test) for all categories. It should be noted that # DataSetA is a completely balanced data set, the training sample part in # DataSetB is a nearly balanced data set, and the validation sample & test sample part in # DataSetB is exactly the same as # DataSetA. The statistics of relevant data sets are shown in Table 1.

Table 1. Statistical information of the experimental dataset.

Dataset

#Training Samples

#Validation Samples

#Test Samples

#Samples Size

#Num_classes

#DataSetA (Original)

50,000 (Original)

5,000 (Original)

10,000 (Original)

65,000 (Original)

10

#DataSetB (Enhanced)

68,259 (Enhanced)

5,000 (Original)

10,000 (Original)

83,259 (Enhanced)

10

Sincere wishes by all authors!

[1] https://pan.baidu.com/s/1vgnrvWBI-u4H16hIOJwXLA?pwd=th7s

Reviewer 2 Report

1. The authors are proposing their models based on the prediction accuracy, however, the accuracy can be influenced by both the positive and negative accuracies. Can the authors also present other metrics such as sensitivity, recall, etc?

2. The abstract does not tell the reader why Chinese news text classification is important. Why, should the reader care about Chinese news text classification? Further, the conclusion in the abstract is not a good enough. Authors may need to improve it. 

3. How is your approach compared to other previous applied approaches? Can you show the parallel results?

4. Data description for this study is hard to follow. Can the authors make it clear? 

5. In the line 111, author have mentioned the two models that have been applied for text classification, however, there are several  other approaches that have not been discussed. Refer the following studies https://www.sciencedirect.com/science/article/abs/pii/S0001457521005042, https://journals.sagepub.com/doi/abs/10.1177/03611981231157393, and https://journals.sagepub.com/doi/full/10.1177/03611981231155187

Author Response

Response to the comments

Dear experts:

First of all, thank you very much for taking the time to help us review the paper and give professional and objective suggestions for revision. I am very happy that our work can be affirmed by you. Next, we will give a point-to-point revised reply to your comments, as follows:

Q1: The authors are proposing their models based on the prediction accuracy, however, the accuracy can be influenced by both the positive and negative accuracies. Can the authors also present other metrics such as sensitivity, recall, etc?

A1:For the description of the metrics, please refer to the description in Section 4.2 of the manuscript. As shown in equations (3)-(6), we also use Precision, Recall, and F1-score metrics to evaluate the performance of our model in addition to accuracy.

Q2:The abstract does not tell the reader why Chinese news text classification is important. Why, should the reader care about Chinese news text classification? Further, the conclusion in the abstract is not a good enough. Authors may need to improve it.

A2:We have been improved the abstract. Abstract: (1) Background: Chinese news text is a popular form of media communication, which can be seen everywhere in China. Chinese news text classification is a key direction of natural language processing (NLP). It is found that the existing deep learning methods need to rely on large-scale tagged corpus in news text classification tasks, and the model is poorly interpretable. (2) Methods: To solve the above problems. This paper proposes a Chinese news text classification method based on key feature enhancement named KFE-CNN. It can effectively expand the semantic information of key features to enhance sample data, and then combine One-hot representation to transform text features into binary vectors and input them into CNN model for training and implementation, thus improving the interpretability of the model and effectively compressing the size of the model. (3) Results: Experimental results show that our method can greatly improve the overall performance of the model, and the average accuracy and F1-score on the THUCNews subset of the public dataset have reached 97.84% and 98%. (4) Conclusions: it fully proved the effectiveness of KFE-CNN method in Chinese news text classification task, and it can also fully demonstrate that key feature enhancement can improve classification performance.

Q3: How is your approach compared to other previous applied approaches? Can you show the parallel results?

A3:By combining the experimental results in Table 4 and Table 5, it can be seen that the overall performance of KFE-CNN model is greatly improved compared with the popular classification methods such as TF-IDF, Naive Bayes, SVM, FastText, LSTM, RNN, TextGCN and CNN.

Q4:Data description for this study is hard to follow. Can the authors make it clear? 

A4: For the description of the data set, please refer to the description in Section 4.1 of the manuscript. The original Chinese dataset #DataSetA used in the experiment are constructed from a subset of THUCNews[1]. The entire dataset covers 10 categories including #Sports, #Finance, #Real Estate, #Home Furnishing, #Education, #Technology, #Fashion, #Current Affairs, #Games, and #Entertainment. The original experimental dataset contains 6,500 (5,000 for train, 500 for validation, and 1,000 for test) samples for each category, and a total number of 65,000 data samples. In addition, a total number of 18,259 new data samples (#Sports 659, #Finance 1,336, #Real Estate 888, #Home Furnishing 4,502, #Education 913, #Technology 1,219, #Fashion 3,393, #Current Affairs 2,307, #Games 1,932, and #Entertainment 1,110) were expanded after data enhancement processing based on the original Training Samples of dataset #DataSetA, and finally a new dataset #DataSetB was obtained, which contained a total number of 83,259 data samples (68,259 for train, 5,000 for validation, and 10,000 for test) for all categories. It should be noted that # DataSetA is a completely balanced data set, the training sample part in # DataSetB is a nearly balanced data set, and the validation sample & test sample part in # DataSetB is exactly the same as # DataSetA. The statistics of relevant data sets are shown in Table 1.

Table 1. Statistical information of the experimental dataset.

Dataset

#Training Samples

#Validation Samples

#Test Samples

#Samples Size

#Num_classes

#DataSetA (Original)

50,000 (Original)

5,000 (Original)

10,000 (Original)

65,000 (Original)

10

#DataSetB (Enhanced)

68,259 (Enhanced)

5,000 (Original)

10,000 (Original)

83,259 (Enhanced)

10

Q5:In the line 111, author have mentioned the two models that have been applied for text classification, however, there are several other approaches that have not been discussed. Refer the following studies https://www.sciencedirect.com/science/article/abs/pii/S0001457521005042, https://journals.sagepub.com/doi/abs/10.1177/03611981231157393, and https://journals.sagepub.com/doi/full/10.1177/03611981231155187

A5:First of all, thank you for recommending three latest documents to us. However, after careful study, it is found that the correlation between the first paper and our research content is not very high, and our article was submitted in February 15th, 2023. However, the online publication dates of the last two documents are March 17th and March 6th, 2023, respectively, and there are conflicts in the dates. I am sorry that we cannot site them normally.

Sincere wishes by all authors!

[1] https://pan.baidu.com/s/1vgnrvWBI-u4H16hIOJwXLA?pwd=th7s

Reviewer 3 Report

1. Some grammatical errors are observed.  Authors should effect corrections. 

2. Authors should indicate the research gap from extant literature that their study is closing. 

3. An analytical table should be created that indicates the available studies in the field and the gap(s) in them should be clearly shown. Also, indication should be made on how the current study fills the gap.

4. The results were not well discussed. How did the study findings compare and/ or contrasts the findings from prior related studies?

5. Authors should indicate the implications and contributions of the study findings to theory and practice. 

6. What are the limitations of the study findings? Authors should note that the directions for further studies are based on these limitations. 

Author Response

Response to the comments

Dear experts:

First of all, thank you very much for taking the time to help us review the paper and give professional and objective suggestions for revision. I am very happy that our work can be affirmed by you. Next, we will give a point-to-point revised reply to your comments, as follows:

Q1:Some grammatical errors are observed.  Authors should effect corrections. 

A1: We have read the full text carefully and corrected the corresponding grammatical errors as much as possible.

Q2:Authors should indicate the research gap from extant literature that their study is closing. 

A2: Combining the experimental results in Table 4, Table 5 and Figure 5, it can be seen that the overall effect of our proposed key feature enhancement (KFE) method after combining CNN(KFE-CNN) and RNN(KFE-RNN) models is 2% higher than the average F1 score of existing CNN and RNN models.

Q3: An analytical table should be created that indicates the available studies in the field and the gap(s) in them should be clearly shown. Also, indication should be made on how the current study fills the gap.

A3: The experimental results in Table 4 can show the performance of the existing popular methods on DataSetA data set. According to the results, it can be seen that CNN and RNN models perform best. However, after integrating the key feature enhancement technology proposed by us, the comprehensive performance of KFE-CNN and KFE-RNN models has been greatly improved. According to the experimental results in Table 5 and Figure 5 (b), the F1 score of our model has increased by 2%, which can fully explain the advantages of KFE for existing models.

Q4:The results were not well discussed. How did the study findings compare and/ or contrasts the findings from prior related studies?

A4: According to the results in Table 4, it can be seen that the comprehensive performance of the CNN model is better than that of the other seven models in the 8 groups of comparative experiments carried out on the original dataset #DataSetA. Its average accuracy value can reach 96.26%, and the min-loss value is 0.13 and the average F1-score on all categories is 96%. The comprehensive performance of RNN is second, with an average F1 score of 94%, and the average F1 scores of other models are between 85% and 93%.  According to the experimental results in Table 5 and Figure 5 (b), the F1 score of KFE-CNN and KFE-RNN model has increased by 2% can reach 98% and 96%, which can fully explain the advantages of KFE for existing models.

Q5: Authors should indicate the implications and contributions of the study findings to theory and practice. 

A5: According to the section introduction, discussion and conclusion, we have given the corresponding core conclusions according to the experimental results. First, it effectively achieves data enhancement by extending the key features, and then combines the One-hot vector to transform the text features into a distributed form and input them into the KFE-CNN model for training and implementation, further improving the interpretability of the model. Second, the experimental results show that our method can not only greatly improve the overall performance of the model, but also reduce the dependence on large-scale manual labeling data and compress the final checkpoint model size. Finally, for long text sample data, the experiment found that partial sample data enhancement is better than full sample enhancement, and the enhanced sample data is universal to different classification models.

Q6:What are the limitations of the study findings? Authors should note that the directions for further studies are based on these limitations. 

A6: We analyzed the limitations of our method in section 4.3 shortages summary, and then we gave a possible direction via AIGC model to improvement KFE step in the future work of section 5.

Sincere wishes by all authors!

Round 2

Reviewer 1 Report

abstract is appropriate

Content of the manuscript is adequate

Methodlogy and result analysis is sufficient to justify the work

References are properly cited

overall manuscript is good

Author Response

Response to the comments

Dear experts:

First of all, thank you very much for taking the time to help us review the paper and give professional and objective suggestions for revision. I am very happy that our work can be affirmed by you. Next, we will give a point-to-point revised reply to your comments, as follows:

Q1:abstract is appropriate

     A: Thank you for your approval and suggestions.

Q2:Content of the manuscript is adequate

     A: Thank you for your approval and suggestions.

Q3:Methodology and result analysis is sufficient to justify the work

     A: Thank you for your approval and suggestions.

Q4:References are properly cited

     A: Thank you for your approval and suggestions.

Q5:overall manuscript is good

     A: Thank you for your approval and suggestions.

Sincere wishes by all authors!

Reviewer 3 Report

1. Authors did not address any of the observations made in the first version of the manuscript. 

2. Authors should proofread their manuscript and correct the grammatical errors observed therein. 

3. The research gap that the current study is filling is lacking. Authors should clearly state their research problem. 

4. Authors should provide a table that analyses and clearly filters out the research gap and problem for the study thus clearly setting out the research agenda for the study. This table will show the inadequacies of prior related studies and how the current study improves or solved the impending problem. 

5. The implications and contributions of the study to theory and practice should be included. 

6. Authors should provide the limitations of the study and also indicate the directions for further studies based on the limitations. 

Author Response

Response to the comments

Dear experts:

First of all, thank you very much for taking the time to help us review the paper and give professional and objective suggestions for revision. I am very happy that our work can be affirmed by you. Next, we will give a point-to-point revised reply to your comments.

Q1:Authors did not address any of the observations made in the first version of the manuscript. 

     A1: As far as possible, we have updated the content and corrected the grammar according to the relevant modification suggestions.

Q2: Authors should proofread their manuscript and correct the grammatical errors observed therein. 

     A2: We have tried our best to read and correct the full text and grammatical errors.

Q3:The research gap that the current study is filling is lacking. Authors should clearly state their research problem. 

     A3: In the second paragraph of the introduction section, we list some shortcomings of the current mainstream text classification methods. Then, in the last paragraph of the introduction section, the main contribution of this paper is given, and the KFE-CNN method is proposed to improve the performance of the text method.

Q4: Authors should provide a table that analyses and clearly filters out the research gap and problem for the study thus clearly setting out the research agenda for the study. This table will show the inadequacies of prior related studies and how the current study improves or solved the impending problem. 

     A4: In order to reveal the differences of these methods more clearly, we make a further summary analysis of the related methods in Table A1 in the Appendix A.

Q5:The implications and contributions of the study to theory and practice should be included. 

     A5: The core contributions and advantages of this method have been described in detail in the introduction, conclusion and discussion of the article. The core principle of this method is introduced in section 3, 3.1, and 3.2, which fully demonstrates the practical process of key feature enhancement.

Q6:Authors should provide the limitations of the study and also indicate the directions for further studies based on the limitations. 

     A6: In the Shortages Summary of subsection 4.5, we describe some limitations of this paper. Then, in the future research work in Section 5, we give some possible improvement directions.

Sincere wishes by all authors!
